# Frame-Level Captions for Long Video Generation with Complex Multi Scenes

## Abstract

Generating long videos that can show complex stories, like movie scenes from scripts, has great promise and offers much more than short clips. However, current methods that use autoregression with diffusion models often struggle because their step-by-step process naturally leads to a serious error accumulation (drift). Also, many existing ways to make long videos focus on single, continuous scenes, making them less useful for stories with many events and changes. This paper introduces a new approach to solve these problems. First, we propose a novel way to annotate datasets at the **frame-level**, providing detailed text guidance needed for making complex, multi-scene long videos. This detailed guidance works with a **Frame-Level Attention Mechanism** to make sure text and video match precisely. In inference, we develop **Parallel Multi-Window Denoising**, a new method that handles a long video as multiple overlapping windows. These windows are processed in parallel, and the noise prediction in overlapping areas is averaged, which allows bidirectional information interaction and introduces no error accumulation. A key feature is that each part (frame) within these windows can be guided by its own distinct text prompt. Our training uses **Diffusion Forcing** to provide the model with the ability to handle time flexibly. We tested our approach on difficult VBench 2.0 benchmarks ("Complex Plots" and "Complex Landscapes") based on the WanX2.1-T2V-1.3B model. The results show our method is better at following instructions in complex, changing scenes and creates high-quality long videos. We plan to share our dataset annotation methods and trained models with the research community.

## 1 Introduction

The ability to create long video sequences from text instructions opens exciting doors for rich, evolving stories, such as turning scripts into videos, producing short films, or showing complex processes. Unlike short clips, long videos provide the needed duration for multiple connected scenes, detailed character interactions, and consistent plotlines that follow complex user requests [17, 42, 32, 23, 15, 20, 22]. However, creating high-quality, consistent, and accurate long videos from text is still a major challenge for current generative models.

A primary difficulty lies in the common autoregressive (step-by-step) methods used with diffusion models to make longer videos. Their sequential way of working naturally leads to errors accumulation over time. This shows up as lower visual quality, the video drifting away from the original text's meaning, and a loss of consistency as the video gets longer, seriously weakening the quality of extended generations [14, 18, 37, 36]. Furthermore, much current research on long videos deals with single, continuous scenes or slowly changing environments. This limited focus reduces the usefulness of long video generation for dynamic stories with many events, which is a key goal for creative

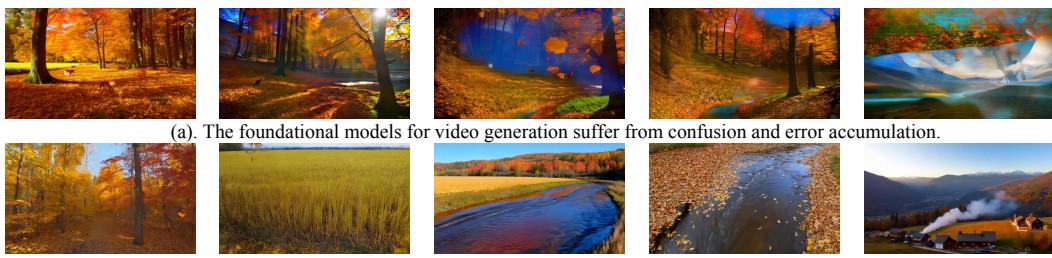

(a). The foundational models for video generation suffer from confusion and error accumulation.

(b). Our method effectively mitigates confusion and error accumulation.

Figure 1: Illustration of issues of error accumulation and semantic confusion in the first row, and videos generated by ours are shown in the second row.

applications. Standard methods using single, general (global) video descriptions struggle with small, quick changes, leading to timing issues [26, 18, 19]. While some approaches try to make multi-shot videos by first detecting scene cuts with tools like PySceneDetect and then processing these shots, as in Presto [30] and Long Context Tuning (LCT) [12], these methods can be complicated, risk losing information, and depend heavily on good shot detection and captioning. They often still describe video at a "shot-text" level, which doesn't fully capture smooth, continuous changes.

To solve these basic problems, our work offers a new way of thinking, focused on highly detailed, frame-level text guidance and a novel non-sequential method for creating the video. Our first main contribution is an innovative **frame-level dataset annotation methodology**. We move beyond general or shot-level captions to provide very detailed text descriptions for each conceptual part (or latent segment) of a video. This rich information about meaning is essential for guiding models to understand and create the complex, changing details needed for stories with many scenes and detailed prompts. This directly addresses the limits of less detailed global or shot-level captions. This detailed annotation is designed to work closely with our **Frame-Level Attention Mechanism**, which clearly links each video segment's visual features to its specific text description, improving content accuracy and consistency over time (Section 4.2).

To properly use such detailed and dynamic text prompts, models need to be trained to handle time flexibly. We achieve this using **Diffusion Forcing** (Section 4.3), a training strategy that shows the model video segments being denoised at different rates. This prepares it to manage varied timing patterns and allows for strong, adaptable inference. Building on these training improvements, we introduce our second major innovation: **Parallel Multi-Window Denoising (PMWD)**, a new inference method designed to create very long videos that are highly consistent (Section 4.4). PMWD divides the target long video into multiple overlapping sections (windows), usually matching the model's training window size. Importantly, unlike step-by-step methods, all these windows are processed *at the same time (in parallel)* during each step of the diffusion denoising process. The data in the overlapping areas between windows is then averaged. This averaging not only ensures smooth connections but also allows information to flow in both directions, meaning later parts of the video can help refine earlier ones. A special feature of PMWD is that each conceptual frame, even within these simultaneously processed windows, can be guided by its own distinct, frame-level prompt.

We test our approach thoroughly using highly challenging benchmarks, specifically the "Complex Plots" and "Complex Landscapes" prompt categories from VBench 2.0 [45]. We use the state-of-the-art open-source WanX2.1-T2V-1.3B model [32] as our base. Our experiments show that our combined frame-level approach is much better at following prompts when creating very long videos with multiple scenes, different characters, and complex actions.

In summary, our main contributions are:

- **Scalable Frame-Level Dataset Methodology:** We introduce an efficient and scalable approach for constructing datasets with dense, frame-by-frame textual annotations. This enables highly granular video-text alignment crucial for generating complex, multi-scene narratives without relying on traditional shot detection.

- **Frame-Level Attention for Precise Guidance:** We propose a novel attention mechanism that directly couples each video segment's visual features with its unique frame-level prompt. This significantly enhances semantic fidelity, content accuracy, and temporal consistency in generated videos.

- **Parallel Multi-Window Denoising for Coherent Long Video Generation:** We develop PMWD, a inference strategy that processes a long video as multiple overlapping windows, denoised simultaneously in parallel. Guided by distinct frame-level prompts and leveraging overlap averaging for bidirectional context, PMWD effectively avoids the error accumulation common in sequential methods. This is enabled by training strategies like Diffusion Forcing that provide temporal flexibility.
- **State-of-the-Art Performance on Complex Videos:** Through comprehensive evaluations on challenging VBench 2.0 benchmarks ("Complex Plots" and "Complex Landscapes") using the WanX2.1-T2V-1.3B model, we demonstrate our integrated approach's superior ability to follow intricate prompts in multi-element long videos, achieving high-fidelity results with minimal error accumulation.

## 2 Related Work

**Video Generation Dataset.** Large-scale video datasets have driven advancements in video generation, but many existing datasets like YouCook2 [47], VATEX [38], and ActivityNet [5] were not designed for this purpose and lack fine-grained annotations. Similarly, large-scale datasets like YTTemporal-180M [43] and HD-VILA-100M [40] suffer from low-quality captions generated through speech recognition, limiting their utility for high-quality video generation. Datasets like Panda-70M [8] offer extensive data but rely on simplistic global descriptions, which hinder the model's ability to capture fine temporal details. Newer datasets, such as Koala-36M [35] and LongTake-HD [41], provide more detailed annotations but still rely on segment-level or shot-based annotations, limiting long-duration video generation. In contrast, our method introduces a frame-level captioning approach, where each frame is independently annotated with a description that maintains contextual relevance to the preceding and succeeding frames. This ensures better alignment between visual content and text while preserving the temporal continuity and motion dynamics, ultimately improving the overall quality of long-form video generation.

**Long Video Generation.** Video generation has evolved from simple single-shot models to more complex long-form and multi-scene models. Early methods relied on GANs [9, 29, 31, 39], constrained by single-domain datasets. Diffusion models [3, 11, 2, 1, 10] introduced temporal layers, enabling motion modeling. DiT-based architectures [4, 25, 28, 17, 42, 32, 23, 15, 20] have achieved tremendous success in scaling diffusion transformers, significantly enhancing video quality. However, these models were limited to generating short clips. FreeNoise [27] and StreamingT2V [13] extended video sequences using auto-regressive methods and temporal attention mechanisms. Gen-L-Video [34] processes videos as sequences of overlapping short clips and employs a temporal co-denoising technique, wherein multiple predictions for each individual frame are averaged. Despite these advancements, challenges in content diversity and temporal consistency persisted. The Diffusion Forcing [6] paradigm addressed these issues by combining diffusion's high-quality generation with auto-regressive models for sequence extension.

In multi-scene video generation, models like Mask2DiT [26], LCT [12], VideoStudio [21], SKYREELS-V2 [7], MovieDreamer [44], StoryAnchors [33], and VGoT [46] focused on scene-level consistency but struggled with temporal coherence across scenes. Recent methods, including StoryDiffusion [48] and MEVG [24], employed attention mechanisms to enhance visual and dynamic consistency. Our approach uses frame-level attention for dynamic scene extension without fixed scene durations, improving flexibility and coherence in long-form videos. Combined with Diffusion Forcing [6], our method ensures smooth scene transitions, extended video lengths, and maintains both visual richness and temporal consistency.

## 3 Frame-Level Dataset

Previous video-text datasets such as Panda70M [8] and Koala-36M [35] provide only global-level captions, resulting in coarse supervision that cannot reflect detailed visual changes within videos. LongTake-HD [41] offers shot-level sub-captions but still depends on explicit shot boundaries, making it difficult to model continuous motion and intra-shot dynamics. In contrast, our dataset uses frame-level uniform sampling and annotation, enabling dense and temporally continuous supervision. This design captures both subtle and significant changes without being limited by artificial segmentation, supporting more precise alignment between video frames and text descriptions. Overall, our dataset

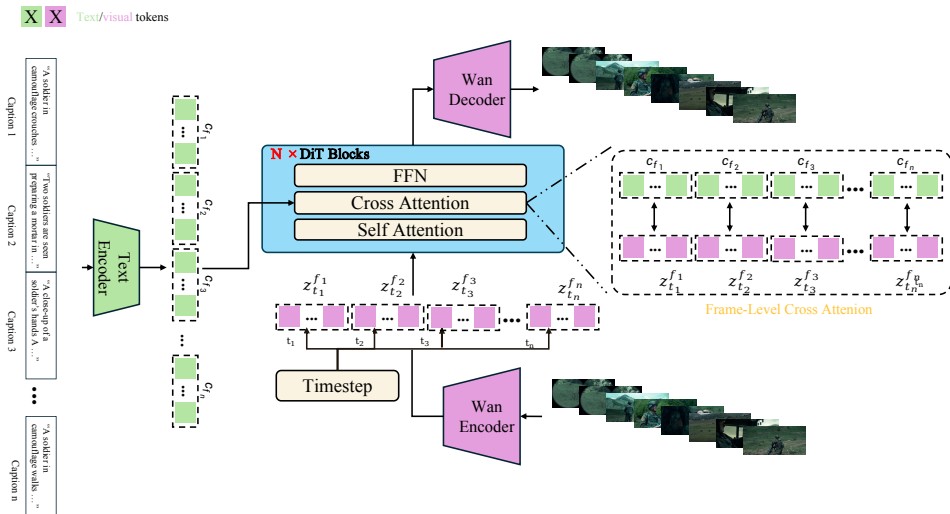

Figure 2: **Overview of the proposed frame-level training method**. Frame-Level Cross-Attention links the visual data of each video segment (latent token) directly to its own specific text description.

offers finer-grained, structurally consistent, and temporally faithful video-text supervision, facilitating improved learning of dynamic visual content.

**Large-scale frame-level video dataset construction.** We present a frame-level video dataset comprising 700,000 high-quality clips, designed to enhance fine-grained text-video alignment and provide dense semantic supervision for diffusion-based video generation models. The dataset systematically balances visual diversity, temporal continuity, and annotation precision, and can be further improved with larger scale in the future. We collect raw videos longer than 10 minutes from multiple platforms, remove near-duplicate content using perceptual hashing, and discard the first and last 10% of frames to ensure the sampled content is dynamic and semantically meaningful. Each processed video is evenly divided into four segments, from which an 8-second continuous clip is extracted. At a frame rate of 24 fps, one frame is sampled every 8 frames, resulting in approximately 24 frames per clip. This design balances temporal context and computational efficiency, and ensures compatibility with mainstream video VAE tokenization schemes, enabling precise one-to-one frame-token supervision.

**Adaptive frame-level annotation.** We utilize multimodal large language models to generate frame-level captions, automatically choosing between shared or independent descriptions based on the degree of visual change. Identical captions are assigned to frames with minimal differences, while significant changes trigger independent frame-level descriptions, achieving unified and adaptive semantic alignment. To enhance consistency and richness, we design structured annotation prompts that require each frame description to cover main subjects, actions, environment, shot size, and camera angle. Outputs strictly follow JSON format with no redundant commentary, ensuring precise semantic and structural alignment at the frame level. We provide the full frame-level annotation prompt, specifying formatting and content requirements to facilitate reproducibility and further research; the template is too long to show here, see appendix. During inference, we use gemini pro 2.5 to convert a user input from short/detailed caption to a frame-level detailed caption. More details can be found in appendix.

## 4 Method

Our work introduces a set of interconnected improvements for DiT-based video diffusion models, aimed at generating complex, long videos. We focus on enhancing how models understand text instructions for each video part, how they handle timing and changes, and how they create long, consistent videos during inference.

## 4.1 Overall Framework

Current Diffusion Transformer (DiT) based models are skilled at creating high-quality short videos. They typically use a VAE (Variational Autoencoder) to compress videos into compact latent data, and a DiT then generates video from these latents. However, when generating long videos with detailed stories and dynamic action, these models face several basic problems:

- **Imprecise Content Control:** Using a single text description (caption) for the entire video often leads to unclear or mixed-up details for different parts of the video, making it hard to accurately control specific events or elements across various scenes.

- **Limited Handling of Timing:** Standard models usually denoise all parts of the video at the same rate in each step. This restricts their ability to show varied motion speeds, different pacing in scenes, or sudden changes effectively.

- **Difficulty with Long Video Coherence:** When creating long videos by generating segments one after another from models trained on short clips, errors tend to build up. This can make the video lose consistency over time.

To address these key challenges, we propose three main contributions:

1. A **Frame-Level Cross-Attention** mechanism (Section 4.2) for precise, localized text-based control over the content of each video segment during training.
2. A **Diffusion Forcing** training strategy (Section 4.3) to teach models how to handle varied timing by exposing them to video segments denoised at different rates.
3. A **Synchronized Multi-Window Denoising (PMWD)** inference method (Section 4.4), designed to generate long, coherent videos by significantly reducing the build-up of errors.

While these principles can be applied to many DiT-based video models, we demonstrate our methods by adapting and fine-tuning the WanX2.1-T2V-1.3B [32] framework, a well-known open-source model, as our base.

## 4.2 Frame-Level Cross Attention

To accurately control video content in line with detailed narratives, we introduce Frame-Level Cross-Attention. This method links the visual data of each video segment (latent token) directly to its own specific text description. Simultaneously, the DiT's standard self-attention mechanism continues to capture overall temporal relationships, ensuring smooth motion. This approach provides both precise local content guidance and global video coherence.

Our process starts by assigning an independent text description to each conceptual "frame" (latent unit) of a video. When the original video is converted into latent data by the VAE, each resulting latent token $z_f$ is directly paired with the embedding of its corresponding frame-level caption, $c_f$. This creates a detailed, one-to-one mapping between text and video segments over time, offering exact guidance for generation. We modify the DiT's cross-attention mechanism so that each latent token $z_f$ attends exclusively to its paired caption embedding $c_f$, rather than to a single caption shared by the entire video. Formally, this is:

$$\text{CrossAttention}(q_f, c_f) = \text{Softmax}\left(\frac{q_f W_q (c_f W_k)^T}{\sqrt{d}}\right)(c_f W_v), \tag{1}$$

where $q_f$ is the query projected from $z_f$, and $W_q, W_k, W_v$ are learnable matrices. This targeted attention mechanism reduces the unclear meaning that can arise from global captions, greatly improving text-to-video alignment and allowing for precise control over dynamic content within each segment.

## 4.3 Diffusion Forcing for Temporal Flexibility

Creating long videos with dynamic action and varied pacing requires the model to handle time flexibly. Standard diffusion models are often limited in this area because they apply the same noise level to all video segments at each step of the denoising process, restricting their ability to generate diverse visual qualities, dynamic changes, or quick scene transitions.

To give models this needed flexibility, we use a **Diffusion Forcing (DF)** strategy during training. This technique assigns an independent noise level to each video segment (latent token) in a training sequence. Specifically, we pick a reference segment, set its target noise removal stage (timestep), and then determine the noise stages for other segments in relation to it: preceding segments get "cleaner" (earlier) timesteps, and subsequent segments get "noisier" (later) timesteps. This approach maintains temporal smoothness while training the model to manage different denoising states simultaneously within one sequence.

This training approach makes the model highly adaptable at inference time. By adjusting a "step-size" parameter—which controls the allowed difference in noise schedules between adjacent segments—we can smoothly shift the generation style. We can opt for fully synchronized diffusion (small step-size, for high consistency) or for more dynamic, evolving outputs (large step-size, resembling autoregressive generation). This adaptability allows the model to produce either smooth, consistent videos or to progressively unfold complex scene transitions and actions as guided by the text prompts. Furthermore, already partially denoised historical segments can serve as stable conditions for generating later segments, aiding long-range consistency without forcing all segments to share the same noise level at the same time.

### 4.4 Flexible Inference Modes for Long Video Generation

The temporal flexibility gained from Diffusion Forcing during training allows for various inference methods to generate videos much longer than the training segments (the "*train short, test long*" approach).

**Sequential Sliding Window Approaches.** Common methods for long video generation use a sequential sliding window. These include simple autoregressive techniques, where a new segment of $M$ latents is generated based on $N - M$ previous latents from an $N$-latent window (often re-noising the context), and more advanced methods like FIFO-Diffusion [16], which uses a queue with diagonally progressing noise levels for better temporal consistency. However, a core problem with all such step-by-step sequential methods is the unavoidable build-up of errors, which reduces quality and long-range consistency in very long videos.

**Parallel Multi-Window Denoising (PMWD).** To effectively overcome this error accumulation problem, we introduce PMWD. This novel inference strategy takes full advantage of our frame-level prompt system to generate long videos more as a complete whole, rather than piece by piece. For a target long video of $L$ latents, we view it as $K$ overlapping windows, each the length of a training segment. Crucially, all $K$ windows are processed *in parallel* (at the same time) during each step of the diffusion denoising process. Every latent, whether new or historical context, is guided by its own dedicated frame-level prompt. This parallel, parallel method for the entire sequence inherently avoids the cascading error build-up seen in typical autoregressive techniques. Latents located in the overlapping regions between adjacent windows are averaged after each denoising step. This averaging, along with the parallel processing, allows information to flow in both directions (bidirectionally) between an earlier and a later window. Unlike methods where only the past influences the future, PMWD allows upcoming video segments to help refine earlier ones. This is especially useful for creating natural-looking scene changes and maintaining consistency in stories with multiple scenes.

## 5 Experimental Results

### 5.1 Experimental Setup

We fully fine-tune the open-source WanX-2.1-T2V-1.3B model with Diffusion Forcing technique on resolution 81x480x832 for 100,000 iterations using our internal dataset (detailed in Section 3) of dense frame-level annotations. Training occurred on H-series GPUs with a global batch size of 64.

### 5.2 Evaluation Dataset

We evaluate the model's capability to generate complex videos by utilizing prompts from the VBench 2.0 benchmark, specifically focusing on the **Complex Plots** and **Complex Landscapes**.

**Complex Plots** assess the model's ability to construct coherent and consistent multi-scene narratives based on prompts describing multi-stage events. These prompts often involve extended descriptions

(150+ words) outlining a sequence of actions or a story with multiple acts, challenging the model to maintain plot consistency and logical flow throughout the generated video.

**Complex Landscapes** evaluate whether the model can faithfully translate long-form landscape descriptions (150+ words) into video, including multiple scene transitions dictated by camera movements. These prompts test the model's understanding of spatial relationships and its ability to render dynamic changes in the environment as described in the text.

## 5.3 Evaluation Metrics

We evaluate video quality using metrics for overall video-text alignment and also propose a new metric for the issue of semantic confusion in multi scenes generation. Let $P_g$ be the global prompt, $V$ the generated video, $\{P_1, \ldots, P_F\}$ the sequence of $F$ frame-level prompts, and $\{V_1, \ldots, V_F\}$ the corresponding sequence of generated frames.

**Standard VBench Evaluation.** To provide a comprehensive assessment of fundamental video quality aspects, particularly for the complex scenarios presented by our chosen VBench 2.0 prompt categories (Complex Plots and Complex Landscapes), we incorporate a curated subset of established metrics from the VBench benchmark. This evaluation focuses on key indicators such as: aesthetic quality, image quality, and motion smoothness. These selected metrics offer standardized measures of the perceptual quality and spatio-temporal coherence of the generated videos.

**Video-Level Video-Text Similarity.** This standard metric evaluates overall coherence between $P_g$ and $V$. $\Phi_V(V)$ represents overall video features (uniformly sample 8 frames as input of ViClip).

$$\mathcal{S}_{\text{global}} = \text{Sim}(\Phi_T(P_g), \Phi_V(V)) \tag{2}$$

where we use a pre-trained vision-language model (e.g., ViCLIP) for text embeddings $\Phi_T(\cdot)$ and video/latent 'frame' embeddings $\Phi_V(\cdot)$, with $\text{Sim}(\cdot, \cdot)$ denoting cosine similarity.

**Confusion Degree (CD).** Despite the widespread use of $\mathcal{S}_{\text{global}}$, this global metric may assign favorable scores even when content from different scenes are inappropriately combined. To pinpoint such temporal and semantic inaccuracies, we introduce the Confusion Degree (CD). A high CD score reveals a model's difficulty in maintaining a clear, sequential narrative, often resulting in a muddled or incoherent visual story. We first define two fundamental frame-level similarity metrics as follows:

$$\begin{aligned} S_{TT}(P_i, P_j) &= \text{Sim}(\Phi_T(P_i), \Phi_T(P_j)) \\ S_{TF}(P_i, V_j) &= \text{Sim}(\Phi_T(P_i), \Phi_V(V_j)) \end{aligned} \tag{3}$$

, where $S_{TT}(P_i, P_j)$ represents **frame-level text-text similarity** and $S_{TF}(P_i, V_j)$ represents **frame-level text-frame similarity**. Then $\tilde{S}_{TT}(P_i, P_j) = S_{TT}(P_i, P_j)/S_{TT}(P_i, P_i)$ and $\tilde{S}_{TF}(P_i, V_j) = S_{TF}(P_i, V_j)/S_{TF}(P_i, V_i)$ are applied as normalization function to ensure $\tilde{S}_{TT}(P_i, P_i) = 1$ and $\tilde{S}_{TF}(P_i, V_i) = 1$.

The confusion degree of a text $P_i$ in the generated video $V$ is defined as:

$$\text{CD}(P_i) = \sum_{j \in \{1, \ldots, F\}} \max(0, \tilde{S}_{TF}(P_i, V_j) - \tilde{S}_{TT}(P_i, P_j)) \tag{4}$$

where $\tilde{S}_{TF}(P_i, V_j) - \tilde{S}_{TT}(P_i, P_j)$ indicates that the content of $P_i$ is more aligned with frame $V_j$ than its inherent semantic relationship with $P_j$ would suggest, thereby signaling confusion. Then the confusion of a video $V$ is defined as

$$\text{CD} = \frac{1}{F} \sum_{i=1}^{F} \text{CD}_i \tag{5}$$

, representing the average confusion degree across all frames. Lower CD values indicate superior narrative consistency and reduced semantic confusion throughout the video.

## 5.4 Comparison and Discussion

**Analysis of Video Generation under Complex Prompts.** Tab. 1 provides a comparative analysis of models trained and inferenced using either global video-level or granular frame-level prompts. When

| Method | Video Length | Prompts Type | Confusion Degree↓ | Video-level Text-Video Consistency↑ | Frame-level Text-Video Consistency↑ | Motion Smoothness↑ | Aesthetic Quality↑ | Image Quality↑ |
|---|---|---|---|---|---|---|---|---|
| DF + Video-level Prompt | 5s | Complex Plot | $0.2952 \pm 0.0461$ | $0.2100 \pm 0.0410$ | $0.1635 \pm 0.0282$ | 98.43 | **59.20** | **67.92** |
| DF + Video-level Prompt | 30s | Complex Plot | $0.2962 \pm 0.0487$ | $0.2053 \pm 0.0368$ | $0.1518 \pm 0.0258$ | **98.63** | 52.02 | 58.07 |
| DF + Frame-level Prompt | 30s | Complex Plot | $\mathbf{0.1385 \pm 0.0498}$ | $\mathbf{0.2196 \pm 0.0309}$ | $\mathbf{0.2054 \pm 0.0231}$ | 98.53 | 55.04 | 61.56 |
| DF + Video-level Prompt | 5s | Complex Landscape | $0.2745 \pm 0.0412$ | $0.2101 \pm 0.0341$ | $0.1831 \pm 0.0227$ | 98.70 | **61.32** | **59.61** |
| DF + Video-level Prompt | 30s | Complex Landscape | $0.2806 \pm 0.0474$ | $0.2066 \pm 0.0351$ | $0.1723 \pm 0.0230$ | 98.58 | 52.63 | 51.02 |
| DF + Frame-level Prompt | 30s | Complex Landscape | $\mathbf{0.1528 \pm 0.0479}$ | $\mathbf{0.2195 \pm 0.0326}$ | $\mathbf{0.2139 \pm 0.0167}$ | **98.99** | 56.31 | 55.98 |

Table 1: Comparing video-level versus frame-level prompting for complex narrative videos. While global Video-Level Text-Video Consistency can yield misleadingly high scores despite internal scene blending or semantic confusion, metrics like Confusion Degree and frame-level consistency more effectively expose these flaws, highlighting the superior prompt adherence of frame-level strategies. Further analysis in Section 5.4.

| Method | Prompt Level | Confusion Degree ↓ | Video-level ↑ Text-Video Consistency | Frame-level ↑ Text-Video Consistency | Motion ↑ Smoothness | Aesthetic ↑ Quality | Image ↑ Quality |
|---|---|---|---|---|---|---|---|
| First-In-First-Out (FIFO) | Video | $0.2962 \pm 0.0487$ | $0.2053 \pm 0.0368$ | $0.1518 \pm 0.0258$ | 98.63 | 52.02 | 58.07 |
| First-In-First-Out (FIFO) | Frame | $0.2416 \pm 0.0514$ | $0.2100 \pm 0.0370$ | $0.1660 \pm 0.0266$ | **98.81** | 51.32 | 59.25 |
| Sequential Sliding Window | Frame | $0.1773 \pm 0.0550$ | $0.2134 \pm 0.0312$ | $0.1842 \pm 0.0227$ | 98.80 | 52.04 | 60.11 |
| Parallel Multi-Window Denoising | Frame | $\mathbf{0.1385 \pm 0.0498}$ | $\mathbf{0.2196 \pm 0.0309}$ | $\mathbf{0.2054 \pm 0.0231}$ | 98.53 | **55.04** | **61.56** |

Table 2: Inference method comparison for 30s complex plot videos. Parallel Multi-Window Denoising (PMWD) achieves lower error accumulation (improved aesthetic/image quality) and better prompt adherence (reduced Confusion Degree, higher text-video consistency) versus causal methods (FIFO, Sliding Window). Detailed analysis in Section 5.4.

generating short videos (e.g., 5 seconds) conditioned on a single **video-level prompt**, the model operates closer to an ideal scenario without temporal error accumulation. However, such prompts often lead to a high Confusion Degree (CD), as the model struggles to render extensive semantic information within a condensed timeframe, resulting in blended or muddled content.

Conversely, employing **frame-level prompts** demonstrates a marked improvement in prompt adherence, evidenced by lower CD scores alongside high frame-level consistency metrics. This enhanced ability to follow detailed, segmented instructions makes the frame-level prompting strategy more reliable and effective for generating coherent multi-scene long videos. Furthermore, metrics such as aesthetic and image quality serve as indirect indicators of error accumulation; significant degradation in these scores over time typically reflects compounding errors. This accumulation is an inherent consequence of the causal nature of sequential generation processes, a fundamental issue that even precise frame-level semantic guidance cannot resolve on its own when operating within such autoregressive frameworks.

**Comparative Analysis of Long Video Inference Strategies.** We further analyze the efficacy of different inference strategies for extending video generation beyond training lengths, comparing our proposed Parallel Multi-Window Denoising (PMWD) with established sequential methods like FIFO-Diffusion and naive sliding windows.

Sequential approaches, by their nature, tackle long video generation segment by segment. Naive sliding window techniques autoregressively generate a new chunk of latents conditioned on a limited history of prior latents (often re-noised to manage error). FIFO-Diffusion [16] offers a more sophisticated sequential mechanism, processing a queue of latents with diagonally increasing noise levels to output one clean latent per step, thereby aiming for better temporal consistency through extended context. While these methods incorporate mechanisms to manage error, such as FIFO's broader context or the re-noising of historical data in naive sliding windows, they fundamentally struggle with the *inevitable accumulation of errors* over very long sequences. This compounding error degrades long-range coherence and overall video quality.

Our proposed **Parallel Multi-Window Denoising (PMWD)** is architecturally designed to overcome this critical limitation. Instead of sequential generation, PMWD processes the entire target long video (composed of multiple overlapping windows) *simultaneously* at each denoising step, with each latent guided by its specific frame-level prompt. This parallel, holistic approach fundamentally disrupts the chain of error propagation seen in sequential methods. The averaging of latents in overlapping regions is a key aspect of PMWD. This, combined with parallel processing, not only fuses information effectively but also transforms the strictly causal dependency of sequential models into a *bidirectional contextual influence*, where information from temporally subsequent windows can refine earlier ones. This capability is particularly advantageous for rendering naturalistic scene transitions and ensuring global narrative consistency.

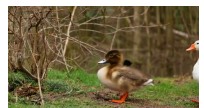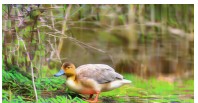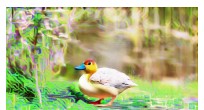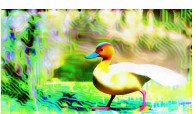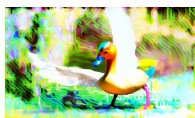

The Ugly Duckling was born into a warm family of ducks but was rejected by the other ducklings due to its unique appearance. It decided to leave home and embark on a journey to find its true place. Throughout the journey, the Ugly Duckling faced many challenges, often feeling lonely and sad. In the harsh winter, it braved the cold and struggled to survive. As spring arrived, the Ugly Duckling discovered that it had transformed into a beautiful swan. Finally finding its true home among other swans, it soared gracefully, becoming the most striking member of the flock. The Ugly Duckling realized that one should never let the opinions of others define them but should trust in the beauty within themselves. The scene is captured in a heartwarming and uplifting style, with soft lighting and a gentle camera movement following the Ugly Duckling's journey from rejection to acceptance and self–discovery.

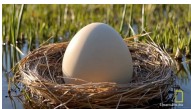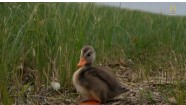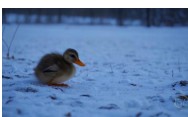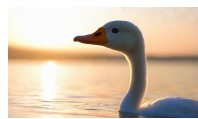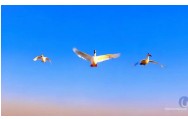

A single, large, greyish egg sits amongst a clutch of smaller, white duck eggs in a cozy nest made of reeds and soft grass, located near a tranquil pond. Soft, warm morning light filters through the reeds. [Extreme Close-Up, Eye-Level]

The same Ugly Duckling sits alone at the edge of the reeds, watching its siblings and mother duck swim and play happily in the pond. A clear sense of sadness is on its face. Soft, warm light. [Medium Shot, Eye-Level, gentle camera slowly pushes in slightly]

The landscape is now covered in a light layer of frost as winter approaches. The same Ugly Duckling shivers, seeking shelter from the cold wind under a bare, thorny bush. Soft, cool, dim light. [Medium Shot, Eye-Level]

Now noticeably larger and more graceful, the young creature, which is the same Ugly Duckling, sees a flock of majestic white swans flying gracefully overhead. It watches them with awe and a deep yearning. Soft, warm, golden hour light. [Medium Shot, Low-Angle, looking up at the sky]

The same swan, now confident and the most radiant of the flock, takes flight with the other swans. They soar gracefully together against a beautiful, clear blue sky, symbolizing its complete self-discovery and happiness. Uplifting music swells. Soft, bright, warm light. [Long Shot, Eye-Level, camera gently tilts upwards following their ascent, ending on a wide shot of them flying freely]

Figure 3: Complex Plot Generation. This figure illustrates the impact of different prompting strategies on visual storytelling performance in a complex narrative task based on The Ugly Duckling. The first row shows results generated using DF with a single global prompt, while the second row presents results from our proposed method that combines DF with multiple tailored prompts (multi-prompting). Our method demonstrates significantly improved coherence, reduced error accumulation, and less narrative confusion across the sequence. The images generated with multi-prompting maintain better stylistic and semantic consistency, showcasing its superiority over the global prompt approach in handling complex plot developments.

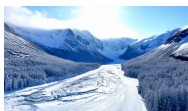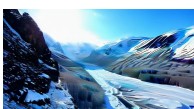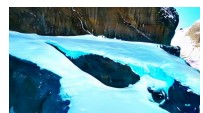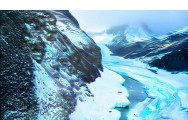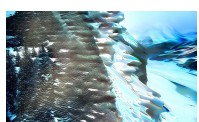

The camera begins with an aerial view of snow–capped mountains, their peaks gleaming with silver under the sunlight. Glaciers wind their way through the landscape, extending into the distance, their icy surfaces reflecting the bright light. A cold wind stirs the air, causing snowflakes to dance in the breeze, the ground sparkling as the snow reflects the intense light. The camera slowly descends into a serene valley, where snow–covered trees stand silent and still, their branches heavy with frost. The camera then moves closer to a rushing glacial river, its waters tumbling over icy rocks, with chunks of ice floating along the surface. As the camera continues its journey, the distant mountain range gradually disappears into the clouds, with only the snow–capped peaks faintly visible through the mist. Finally, the camera pulls back, revealing the vastness of the snow–covered landscape, where the ice and sky merge into a frozen, serene vista. The scene is captured in a realistic, nature documentary style, with a focus on the beauty and tranquility of the winter landscape.

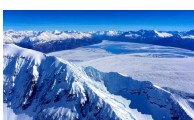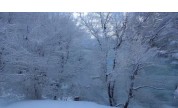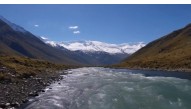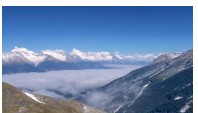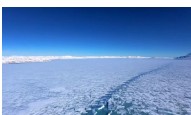

The camera begins with a majestic aerial view of snow-capped mountains, their peaks gleaming with silver under the brilliant sunlight. [Bird's Eye View, Long Shot]

The camera gently drifts past the frost-laden branches of the same snow-covered trees, emphasizing their still and silent presence in the serene valley. [Close-Up on branches, Eye-Level, slow drift]

As the camera continues its journey alongside the river, the distant mountain range becomes a more prominent feature in the background, still snow-capped and gleaming. [Long Shot, Eye-Level, camera panning slightly up towards mountains]

The camera holds on the view of the distant mountain range, which gradually begins to be touched by wisps of incoming clouds. The peaks are still clearly visible. [Long Shot, Eye-Level, static]

The camera pulls back to its final position, a breathtaking high-angle view of the vast snow-covered landscape, where the ice and sky merge completely into a frozen, serene, and tranquil vista. Realistic, nature documentary style. [Extreme Long Shot, High-Angle, static]

Figure 4: Complex Landscapes Generation. This figure compares two prompting strategies for generating complex scenes. The top row uses DF + global prompt, while the bottom row shows results from our method: DF + multi-prompt. Our approach significantly reduces content drift and error accumulation across frames. By using multiple prompts tailored to each scene segment, it achieves higher accuracy and coherence, capturing the complexity and progression of the winter landscape more effectively than the global prompt method.

## 6   Conclusion

Generating long, narratively complex videos with high fidelity remains challenging, primarily due to issues with coarse semantic guidance and the error accumulation inherent in common sequential generation techniques. We propose a comprehensive solution combining fine-grained frame-level annotations, novel training strategies, and a Parallel Multi-Window Denoising (PMWD) inference method. Our experiments on demanding VBench 2.0 benchmarks demonstrate that this integrated system significantly improves prompt adherence for complex, multi-scene narratives in ultra-long videos, achieving high-quality results with minimal error accumulation.

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
