# OpenReview forum: "Frame-Level Captions for Long Video Generation with Complex Multi Scenes"
_NeurIPS.cc/2025/Conference — Submitted to NeurIPS 2025_

### Official Review · Reviewer_HnVY · 2025-07-03

**Clarity:** 2
**Significance:** 2
**Originality:** 2
**Rating:** 2
**Confidence:** 4

**Summary:**

This paper addresses the challenges of semantic confusion and error accumulation in generating long videos. The authors propose a framework that begins with creating a dataset with dense, frame-level text annotations. To leverage this data, they introduce a Frame-Level Attention mechanism for precise text-to-video alignment and use a Diffusion Forcing training strategy to give the model temporal flexibility. The non-sequential inference method, Parallel Multi-Window Denoising (PMWD), processes the entire video in parallel, with overlapping windows to prevent error buildup.

**Questions:**

see above

**Ethical Concerns:**

["NO or VERY MINOR ethics concerns only"]

**Final Justification:**

Considering the comments from other reviewers and the author's rebuttal, I will maintain my original rating for this paper.

**Limitations:**

see above

**Quality:**

4

**Strengths And Weaknesses:**

**Strengths**
1. The authors tackle the challenge of long video generation from a novel perspective, introducing a fresh line of thought for the research community.
2. The figures and tables are clear and effectively aid the reader in understanding the methodology and results.

**Weaknesses**
1. The technical contribution is limited. The dataset annotations are derived by prompting an existing MLLM, and the diffusion forcing strategy is directly borrowed from prior work [1]. Additionally, the proposed PMWD method is identical to temporal MultiDiffusion [2], which is a widely adopted strategy in previous long video generation studies [3, 4].
2. The paper's presentation is confusing. The problem setting (e.g., whether it focuses on single-shot long video generation or multi-shot video generation) is not clearly discussed. The comparison settings in Figures 3 and 4 are also unclear.
3. The paper lacks any ablation experiments to validate its design.

[1] Diffusion forcing: Next-token prediction meets full-sequence diffusion

[2] Multidiffusion: Fusing diffusion paths for controlled image generation

[3] Avid: Any-length video inpainting with diffusion model

[4] Gen-l-video: Multi-text to long video generation via temporal co-denoising

---

> ### Author Rebuttal · Authors · 2025-07-31
>
> ## General Response to All Reviewers:
>
> 1.
>    **The first trial of the frame-level caption method**, representing a significant attempt in dataset annotation paradigms. This approach offers unique advantages:
>
>    - First, the annotation method is simple, requiring only uniform frame sampling without considering clip, shot, or scene boundaries. This greatly reduces annotation difficulty while capturing natural transitions in the dataset, serving as a crucial technique for advancing multi-shot/scene generation scenarios.
>    - Second, unlike direct frame-level image caption strategies, our core innovation involves using state-of-the-art Vision-Language Models (VLM) to first establish global understanding of multi-shot long videos before annotating individual frames. This fundamentally differs from traditional pipelines that process and caption single clips separately, providing substantial significance.
>    - Finally, fine-grained frame-level captions can be flexibly rewritten and summarized into clip/chunk/shot/scene/video-level captions using LLM models based on specific usage scenarios. Particularly in chunk-level autoregressive inference, frame-level captions can form chunk prompts of arbitrary lengths without concerns about shot or scene boundaries—an issue that cannot be resolved using shot-level or scene-level captions directly.
> 2.
>    **This paper introduces semantic confusion in multi shot autoregressive long video scenarios**, going beyond the commonly discussed error accumulation problem. More text descriptions do not improve generation quality; instead, they cause semantic confusion during generation. This occurs because each chunk requires its own dedicated prompt rather than a global video-level prompt—a critical insight not addressed in previous autoregressive works focused primarily on simple actions and single shots. To quantify this issue, we introduce the 'confusion degree' metric.
> 3.
>    This paper also **addresses how to effectively utilize frame-level captions** through our proposed Frame-Level Attention Mechanism. By restricting each frame/chunk/shot to attend only to its corresponding prompt while maintaining global information interaction through visual self-attention, we find this approach can still achieve overall understanding.
>    This concept demonstrates remarkable foresight, as evidenced by its subsequent adoption in Seedance—the current SOTA closed-source video generation model released after our work. Seedance employs a similar mechanism where multi-modal visual-text attention operates exclusively within shots, while global information interaction through temporal self-attention (visual only) prevents inter-shot semantic confusion. This convergence confirms our approach's forward-looking and heuristic value, particularly for future advancements in multi-shot/scene long video generation.
> 4.
>    **Regarding our proposed PMWD**, we emphasize it represents just one optional inference method after frame-level training—not a core contribution of this paper. While the original PMWD shows significant advantages in addressing error accumulation, simpler diffusion-forcing sliding window inference methods could also be viable if they solve the error accumulation issue (e.g., using distribution matching techniques like DMD used in Causvid or self-forcing). Notably, PMWD's inherent parallelization capability provides distinct advantages with sufficient GPU resources, and if combined with compressed representations of infinite historical frames, offers substantial future potential.
> 5.
>    **Open Source Commitment**: We will promptly release our dataset processing code, model training/inference/evaluation code, and part of multi-shot video data with frame-level caption annotations.
>
> 6. **More clear implementation details:**
>     As described in line 297 and the caption of Table 1, we present a comparative analysis of models trained on the Wanx2.1 dataset using either global video-level or frame-level prompts under identical training and inference configurations. All experiments were conducted using our proposed Parallel Multi-Window Denoising (PMWD) method. A 5-second video corresponds to 21 latents, which matches the frame length used during Wanx training; therefore, we follow diffusion forcing protocols by treating these 21 latents as a sliding window unit. After each generation step, the window slides by 1/3 of its length (7 latents). For the FIFO approach, we adhere to the original paper's configuration with a slide increment of 1 latent per step. While these implementation details do not affect the conclusions, we apologize for not clearly presenting them. We will provide more detailed explanations in the main text and supplementary materials according to the reviewers' suggestions, and ensure reproducibility through open-source code.
>
> Finally, we sincerely thank all reviewers for their meticulous reviews and kindly request consideration of the heuristic contributions this paper offers to the field.
>
> ## Response To Reviewer uVip
>
> 1. The technical contribution is limited
>
> We are the first to propose frame-level captioning within the comprehensive understanding of entire videos rather than isolated video clips, leveraging the robust comprehension capabilities of Vision-Language Models (VLMs). The key distinctions and significance have been detailed in our general response.
> Notably, Temporal MultiDiffusion[2] cannot generate specific chunk-level prompts without frame-level annotations.
>
> 2.  The problem setting is confusing.
>
> Our paper title clearly states "Long Video Generation with Complex Multi Scenes", and we have repeatedly explained the multi-shot long video scenarios throughout the manuscript. Furthermore, our test set incorporates complex narrative plots and multi-shot scenes. However, we will further refine our paper based on your valuable feedback. We sincerely apologize for any confusion caused and will thoroughly address this issue in subsequent revisions.
>
> 3. The paper lacks any ablation experiments to validate its design.
>
> In Table 1, we systematically validate the effectiveness of frame-level prompts versus video-level prompts in mitigating semantic confusion while maintaining comparable quality levels.
> In Table 2, we demonstrate the efficacy of PMWD in addressing error accumulation during autoregressive long video generation through direct comparisons with state-of-the-art methods including Diffusion Forcing and FIFO.

---

### Official Review · Reviewer_r4uF · 2025-07-03

**Clarity:** 4
**Significance:** 3
**Originality:** 4
**Rating:** 5
**Confidence:** 4

**Summary:**

The paper provide frame-level annotated dataset and corresponding long-video generation models, together with diffusion forcing and PMWD strategy for long video generation. It overcomes shared deficiency of current baseline models including temporal accumulated errors and struggling of complex scenes, and limited handling of timing. The results shows better quality on Vbench ("Complex Plots" and "Complex Landscapes"). Great job.

**Questions:**

See weakness part, total 4 questions and detailed actions are provided.

Also, the release of the dataset significantly influences the broad contribution of this paper. The community is looking at you :-)

**Ethical Concerns:**

["NO or VERY MINOR ethics concerns only"]

**Final Justification:**

Good paper, I will keep my score.

**Limitations:**

yes

**Quality:**

4

**Strengths And Weaknesses:**

Strengths

1. The proposed frame-level video dataset will contribute a lot to the community. It is first-of-a-kind in this large scale.
2. Frame-level attention, and diffusion forcing helps the consistency and necessary dynamics of complex plot and scene applications.
3. PMWD introduce a balanced strategy over the coherence for long video generation.

Weakness

1. The paper shows plausible quality on complex long video generation with dynamic scenes. But it is still necessary to include the base performance of general T2V performance by only global text prompt on the entire Vbench benchmark. How does the introduction of frame-level dataset & attention influence the foundational ability of the original backbone model (Wan2.1)?

2. The quantitative evaluation is lack of competitors. It is necessary to include several representative open/close-source model into the comparison (HunyuanVideo, Cogvideo, Kling, Pika, Veo3, etc.).

3. The therotical/practical insight of comparing frame-level attention VS video-level attention is suggested. What does this new fine-grained dataset and approach modify the pretained representation/distribution of the original backbone model? For example, the comparison of the token-level attention mapping and according discussion will be very helpful.

4. The dataset is mainly annotated by Gemini. But the quality of the annotation itself is lack of manual validation. How good the quality of the dataset is crucial. Even a portion of evaluation is suggested.

---

> ### Author Rebuttal · Authors · 2025-07-31
>
> ## General Response to All Reviewers:
>
> 1.
>    **The first trial of the frame-level caption method**, representing a significant attempt in dataset annotation paradigms. This approach offers unique advantages:
>
>    - First, the annotation method is simple, requiring only uniform frame sampling without considering clip, shot, or scene boundaries. This greatly reduces annotation difficulty while capturing natural transitions in the dataset, serving as a crucial technique for advancing multi-shot/scene generation scenarios.
>    - Second, unlike direct frame-level image caption strategies, our core innovation involves using state-of-the-art Vision-Language Models (VLM) to first establish global understanding of multi-shot long videos before annotating individual frames. This fundamentally differs from traditional pipelines that process and caption single clips separately, providing substantial significance.
>    - Finally, fine-grained frame-level captions can be flexibly rewritten and summarized into clip/chunk/shot/scene/video-level captions using LLM models based on specific usage scenarios. Particularly in chunk-level autoregressive inference, frame-level captions can form chunk prompts of arbitrary lengths without concerns about shot or scene boundaries—an issue that cannot be resolved using shot-level or scene-level captions directly.
> 2.
>    **This paper introduces semantic confusion in multi shot autoregressive long video scenarios**, going beyond the commonly discussed error accumulation problem. More text descriptions do not improve generation quality; instead, they cause semantic confusion during generation. This occurs because each chunk requires its own dedicated prompt rather than a global video-level prompt—a critical insight not addressed in previous autoregressive works focused primarily on simple actions and single shots. To quantify this issue, we introduce the 'confusion degree' metric.
> 3.
>    This paper also **addresses how to effectively utilize frame-level captions** through our proposed Frame-Level Attention Mechanism. By restricting each frame/chunk/shot to attend only to its corresponding prompt while maintaining global information interaction through visual self-attention, we find this approach can still achieve overall understanding.
>    This concept demonstrates remarkable foresight, as evidenced by its subsequent adoption in Seedance—the current SOTA closed-source video generation model released after our work. Seedance employs a similar mechanism where multi-modal visual-text attention operates exclusively within shots, while global information interaction through temporal self-attention (visual only) prevents inter-shot semantic confusion. This convergence confirms our approach's forward-looking and heuristic value, particularly for future advancements in multi-shot/scene long video generation.
> 4.
>    **Regarding our proposed PMWD**, we emphasize it represents just one optional inference method after frame-level training—not a core contribution of this paper. While the original PMWD shows significant advantages in addressing error accumulation, simpler diffusion-forcing sliding window inference methods could also be viable if they solve the error accumulation issue (e.g., using distribution matching techniques like DMD used in Causvid or self-forcing). Notably, PMWD's inherent parallelization capability provides distinct advantages with sufficient GPU resources, and if combined with compressed representations of infinite historical frames, offers substantial future potential.
> 5.
>    **Open Source Commitment**: We will promptly release our dataset processing code, model training/inference/evaluation code, and part of multi-shot video data with frame-level caption annotations.
> 6. **More clear implementation details:**
>     As described in line 297 and the caption of Table 1, we present a comparative analysis of models trained on the Wanx2.1 dataset using either global video-level or frame-level prompts under identical training and inference configurations. All experiments were conducted using our proposed Parallel Multi-Window Denoising (PMWD) method. A 5-second video corresponds to 21 latents, which matches the frame length used during Wanx training; therefore, we follow diffusion forcing protocols by treating these 21 latents as a sliding window unit. After each generation step, the window slides by 1/3 of its length (7 latents). For the FIFO approach, we adhere to the original paper's configuration with a slide increment of 1 latent per step. While these implementation details do not affect the conclusions, we apologize for not clearly presenting them. We will provide more detailed explanations in the main text and supplementary materials according to the reviewers' suggestions, and ensure reproducibility through open-source code.
>
> Finally, we sincerely thank all reviewers for their meticulous reviews and kindly request consideration of the heuristic contributions this paper offers to the field.
>
> ## Response To Reviewer r4uF
>
> 1. The paper shows plausible quality on complex long video generation with dynamic scenes. But it is still necessary to include the base performance of general T2V performance by only global text prompt on the entire Vbench benchmark. How does the introduction of frame-level dataset & attention influence the foundational ability of the original backbone model (Wan2.1)?
>
> We also show the Vbench quality result on 5s video in Tab. 1 and Tab. 2.
> The introduction of frame-level dataset & attention does not influence the foundational ability of the original backbone model (Wan2.1). Through testing on other metrics in Vbench 1.0 and 2.0, performance is more influenced by the learning rate during the finetuning stage and the quality of the training dataset itself. While we collected a large number of datasets, as this is research work, we did not carefully select high-quality data. As a result, after training on our dataset, both video-level prompts and frame-level prompts experienced some performance loss, but they remain comparable when compared directly.
>
>
> 2. The release of the dataset significantly influences the broad contribution of this paper.
>
>  We will promptly release our dataset processing code, model training/inference/evaluation code, and high-quality single and multi shot video data with frame-level caption annotations.
>
>
> 3. The therotical/practical insight of comparing frame-level attention VS video-level attention is suggested. What does this new fine-grained dataset and approach modify the pretained representation/distribution of the original backbone model? For example, the comparison of the token-level attention mapping and according discussion will be very helpful
>
> This is an excellent suggestion. However, our analysis indicates that the impact is not significant. The functionality of each attention head remains largely unchanged, with some heads still responsible for global information and others for local information. In the text-visual cross-attention component, there is a noticeable hard mask effect, but this is primarily due to our implementation rather than being learned.
> Since we cannot include images here, we will present the experiments in subsequent supplementary materials.
>
>
> 4. The dataset is mainly annotated by Gemini. But the quality of the annotation itself is lack of manual validation. How good the quality of the dataset is crucial. Even a portion of evaluation is suggested.
>
> Even with the most advanced Gemini Pro, its comprehension ability is still far below that of humans. We will include a detailed analysis of this aspect in the supplementary materials.

---

> ### Comment · Reviewer_r4uF · 2025-08-06
> **Thanks for the rebuttal content.**
>
> Good paper, I will keep my score.

---

### Official Review · Reviewer_Tfgw · 2025-07-03

**Clarity:** 2
**Significance:** 3
**Originality:** 2
**Rating:** 3
**Confidence:** 4

**Summary:**

This paper proposes a pipeline for multi-scene long video generation, including data annotation, model design and inference strategy. For data, a 700k video dataset with frame-level captioning is provided. For method, it modifies the cross attention layer of WanX2.1 model, allowing each latent exclusively attends to the paired frame-level caption, enabling context change in long videos. The Diffusion Forcing strategy is utilized in training stage. At inference, instead of using auto-regressive generation, this paper proposes to simultaneously denoise multiple overlapping time windows in parallel to eliminate error accumulation. Ablation studies are conducted on VBench 2.0 to validate the effectiveness of the proposed pipeline.

**Questions:**

1. Comparison with baseline long video generation methods should be included.

2. What is the actual annotation pipeline? How is "adaptive" frame-level annotation performed? Please give further explanation.

3. Why do you use DF training when the inference only handles unified noise level per step?

4. Why are there temporal jittering in the generation results? Is it a drawback of the PMWD inference strategy?

**Ethical Concerns:**

["NO or VERY MINOR ethics concerns only"]

**Final Justification:**

As clarified in the rebuttal, DF training is only used for comparison, and PMWD is not a core contribution of the paper. As such, the main contribution seems to be the use of frame-level prompts for long video generation, which is rather trivial, especially given the simplicity of the "adaptive frame-level annotation." My concerns about the writing and temporal jittering are also not fully addressed. Therefore, I will keep my score as a borderline reject.

**Limitations:**

yes

**Quality:**

2

**Strengths And Weaknesses:**

Strengths:

1. The proposed multi-prompt + PMWD inference strategy is simple yet effective, exhibiting superior quantitative and qualitative performance on complex video generation compared to the autoregressive variants.

2. A video dataset with frame-level dense captioning is provided, which is a valuable contribution to the community.


Weakness:

1. The paper is not well-structured and is confusing or unclear in several parts. For example, in Figures 3–4 of the manuscript, the inference strategies used in each row are not indicated. Is the first row using FIFO and the second using PMWD, similar to Figure 4-6 in the appendix? Additionally, none of the figures are referenced or discussed in the main text.

2. No baseline comparison experiments are conducted beyond the ablation variants. The paper does mention works with similar tasks, such as LCT, Mask2DiT, but none of them are qualitatively or quantitatively compared.

3. The annotation pipeline described in the manuscript contradicts the prompt provided in the appendix. Specifically, the claimed strategy of "automatically choosing between shared or independent descriptions based on the degree of visual change" in "Adaptive frame-level annotation" never occurs in the provided Gemini prompt. The actual annotation strategy appears to be simply sampling frames evenly and annotating each frame with Gemini, which is not adaptive at all.

4. The choice of Diffusion Forcing training strategy is confusing. DF is designed to handle varying noise levels within a sequence to enable autoregressive inference, but since the proposed PMWD inference strategy only denoises tokens at the same noise level per step, I do not see the rationale for using DF training strategy.

5. As suggested in the supplementary videos, the PMWD strategy seems to cause jittering at the boundaries between time windows.

---

> ### Author Rebuttal · Authors · 2025-07-31
>
> ## General Response to All Reviewers:
>
> 1.
>    **The first trial of the frame-level caption method**, representing a significant attempt in dataset annotation paradigms. This approach offers unique advantages:
>
>    - First, the annotation method is simple, requiring only uniform frame sampling without considering clip, shot, or scene boundaries. This greatly reduces annotation difficulty while capturing natural transitions in the dataset, serving as a crucial technique for advancing multi-shot/scene generation scenarios.
>    - Second, unlike direct frame-level image caption strategies, our core innovation involves using state-of-the-art Vision-Language Models (VLM) to first establish global understanding of multi-shot long videos before annotating individual frames. This fundamentally differs from traditional pipelines that process and caption single clips separately, providing substantial significance.
>    - Finally, fine-grained frame-level captions can be flexibly rewritten and summarized into clip/chunk/shot/scene/video-level captions using LLM models based on specific usage scenarios. Particularly in chunk-level autoregressive inference, frame-level captions can form chunk prompts of arbitrary lengths without concerns about shot or scene boundaries—an issue that cannot be resolved using shot-level or scene-level captions directly.
> 2.
>    **This paper introduces semantic confusion in multi shot autoregressive long video scenarios**, going beyond the commonly discussed error accumulation problem. More text descriptions do not improve generation quality; instead, they cause semantic confusion during generation. This occurs because each chunk requires its own dedicated prompt rather than a global video-level prompt—a critical insight not addressed in previous autoregressive works focused primarily on simple actions and single shots. To quantify this issue, we introduce the 'confusion degree' metric.
> 3.
>    This paper also **addresses how to effectively utilize frame-level captions** through our proposed Frame-Level Attention Mechanism. By restricting each frame/chunk/shot to attend only to its corresponding prompt while maintaining global information interaction through visual self-attention, we find this approach can still achieve overall understanding.
>    This concept demonstrates remarkable foresight, as evidenced by its subsequent adoption in Seedance—the current SOTA closed-source video generation model released after our work. Seedance employs a similar mechanism where multi-modal visual-text attention operates exclusively within shots, while global information interaction through temporal self-attention (visual only) prevents inter-shot semantic confusion. This convergence confirms our approach's forward-looking and heuristic value, particularly for future advancements in multi-shot/scene long video generation.
> 4.
>    **Regarding our proposed PMWD**, we emphasize it represents just one optional inference method after frame-level training—not a core contribution of this paper. While the original PMWD shows significant advantages in addressing error accumulation, simpler diffusion-forcing sliding window inference methods could also be viable if they solve the error accumulation issue (e.g., using distribution matching techniques like DMD used in Causvid or self-forcing). Notably, PMWD's inherent parallelization capability provides distinct advantages with sufficient GPU resources, and if combined with compressed representations of infinite historical frames, offers substantial future potential.
> 5.
>    **Open Source Commitment**: We will promptly release our dataset processing code, model training/inference/evaluation code, and part of multi-shot video data with frame-level caption annotations.
> 6. **More clear implementation details:**
>     As described in line 297 and the caption of Table 1, we present a comparative analysis of models trained on the Wanx2.1 dataset using either global video-level or frame-level prompts under identical training and inference configurations. All experiments were conducted using our proposed Parallel Multi-Window Denoising (PMWD) method. A 5-second video corresponds to 21 latents, which matches the frame length used during Wanx training; therefore, we follow diffusion forcing protocols by treating these 21 latents as a sliding window unit. After each generation step, the window slides by 1/3 of its length (7 latents). For the FIFO approach, we adhere to the original paper's configuration with a slide increment of 1 latent per step. While these implementation details do not affect the conclusions, we apologize for not clearly presenting them. We will provide more detailed explanations in the main text and supplementary materials according to the reviewers' suggestions, and ensure reproducibility through open-source code.
>
> Finally, we sincerely thank all reviewers for their meticulous reviews and kindly request consideration of the heuristic contributions this paper offers to the field.
>
>
> ## Response To Reviewer Tfgw
>
> 1. Further explanation for "automatically choosing between shared or independent descriptions based on the degree of visual change" and "adaptive frame-level annotation".
>
> Unlike direct frame-level image caption strategies, our core innovation involves using state-of-the-art Vision-Language Models (VLM) to first establish global understanding of multi-shot long videos before annotating individual frames by providing all the keyframes evenly sampled from the entire video. A video may contain shot transitions, scene changes, significant action variations, or subtle modifications. Our concept of "adaptive or automatically" refers to the VLM's capability to recognize these scenarios and generate appropriate frame-level annotations. We apologize for the lack of clarity in our original presentation and will revise this explanation in the main text according to your suggestions.
>
> 2. Why do you use DF training when the inference only handles unified noise level per step?
>
> The focus of this paper is on the impact of video-level prompts and frame-level prompts on the model, so the main difference during training is whether frame-level prompts are used. However, since our test scenario involves multi-shot autoregressive long videos, we modified the video-level timestep flow matching to diffusion forcing's frame-level timestep.
> This modification enables direct comparison with the two state-of-the-art approaches— sliding window inference of diffusion forcing and FIFO inference. Notably, even though PMWD utilizes video-level timesteps, diffusion forcing remains compatible with this configuration after training.
>
> 3. Comparison with baseline long video generation methods should be included.
> The field of autoregressive long video generation is extremely cutting-edge, and fair comparisons are challenging due to open-source and resource constraints.
>  The LCT and Mask2DiT methods you mentioned are both ByteDance works that have not been open-sourced. In particular, different research groups utilize varying datasets and training schemes, making direct comparisons difficult.
>  Therefore, we reproduced diffusion forcing training using the same internal data and identical settings, and compared against the most mainstream approaches of sliding window and FIFO. We also experimented with the history noise scheme from diffusion forcing v2, but observed negligible performance differences.
>   Our experiments have clearly and fairly validated our conclusions regarding frame-level semantic confusion and the effect of frame-level caption.
>
> 4. Why are there temporal jittering in the generation results? Is it a drawback of the PMWD inference strategy?
>
> Temporal jittering occurs because sliding window-based methods only have 21 latents, resulting in a very short visible range. Even with overlap, the maximum overlap is 20 latents. As a training-free method, PMWD solves the problem of error accumulation but still has issues with consistency and temporal jittering. While this is indeed a drawback of the current most basic PMWD implementation, the method has significant potential with its parallel characteristics. It will have greater room for improvement if combined with longer historical context and framepack context compression.
>
> 5. Further explanation for "automatically choosing between shared or independent descriptions based on the degree of visual change" and "adaptive frame-level annotation".
>
> Unlike direct frame-level image caption strategies, our core innovation involves using state-of-the-art Vision-Language Models (VLM) to first establish global understanding of multi-shot long videos before annotating individual frames by providing all the keyframes evenly sampled from the entire video. A video may contain shot transitions, scene changes, significant action variations, or subtle modifications. Our concept of "adaptive or automatically" refers to the VLM's capability to recognize these scenarios and generate appropriate frame-level annotations. We apologize for the lack of clarity in our original presentation and will revise this explanation in the main text according to your suggestions.

---

> > ### Comment · Reviewer_Tfgw · 2025-08-05
> >
> > Thanks to the authors for their reply. As clarified in the rebuttal, DF training is only used for comparison, and PMWD is not a core contribution of the paper. As such, the main contribution seems to be the use of frame-level prompts for long video generation, which is rather trivial, especially given the simplicity of the "adaptive frame-level annotation."
> >
> > My concerns about the writing and temporal jittering are also not fully addressed. Therefore, I will keep my score as a borderline reject.

---

### Official Review · Reviewer_uVip · 2025-07-04

**Clarity:** 2
**Significance:** 2
**Originality:** 3
**Rating:** 3
**Confidence:** 4

**Summary:**

The proposed paper is oriented on solving the challenge of generating long, multi-scene videos by text prompts, where existing autoregressive diffusion models suffer from error accumulation and limited scene diversity. The authors propose three key innovations: first — a frame-level dataset annotation methodology that assigns detailed text descriptions to individual frames or latents using multimodal LLMs, enabling precise video-text alignment, second — a frame-level cross-attention mechanism that binds each video segment’s visual features to its specific text prompt during training, improving semantic fidelity and third — a parallel Multi-Window Denoising inference strategy that processes overlapping video frames in parallel, evaluating average predictions in overlapping regions to enable bidirectional context flow and eliminate sequential error drift. The approach is based at the top of the WanX2.1-T2V-1.3B model and further evaluated on VBench 2.0’s Complex Plots and Complex Landscapes benchmarks. The reported results show significant improvements in prompt adherence, temporal consistency, and reduced semantic confusion  in comparison to global-prompt baselines and sequential methods like FIFO-Diffusion.

**Questions:**

The following actions may improve the paper
1. Computational cost evaluation experiments
2. Provide clear description of the strategy to chose reference segment in DiffusionForcing and other implementation details, especially when you mention to assign 'cleaner' (earlier) timesteps to past segments and 'noisier' (later) to future segments.
3. Why only Gemini Pro 2.5 was used for prompt generation. How will the quality metrics value change with other VLM use?
4. Is there any human interpretation of Confusion Degree?
5. Can we use PMWD for non-overlapping frame shifts? What will be the effect of such operations?

**Ethical Concerns:**

["NO or VERY MINOR ethics concerns only"]

**Final Justification:**

I will remain my score according to the authors’ answers

**Limitations:**

Yes

**Paper Formatting Concerns:**

No paper formatting concerns detected

**Quality:**

3

**Strengths And Weaknesses:**

Strengths.

The paper is technically rigorous. Experiments are comprehensive, using challenging Complex Plots and Landscapes VBench 2.0 categories and introducing a novel metric Confusion Degree to evaluate semantic drift. The results show statistically significant gains in text and video alignment and reduced confusion degree in comparison with the baselines. The proposed approach combines diffusion’s high-quality generation with auto-regressive models for sequence extension and as a result ensures smooth scene transitions, extended video lengths, and maintains both visual richness and temporal consistency.

Weaknesses.

PMWD in terms of parallel processing of multiple windows may lead to high memory/FLOPs overhead, which seems to be underexplored in this paper. If we speak about long videos (1+ minutes and more) there should be some architectural scalability limits which were not properly discussed and may lead to a limited use. At the experimental stage the authors focus on two challenging VBench categories, whereas testing on broader domains (e.g., human actions, DIY operations) would strengthen the claims. The frame-level annotation pipeline relies on Gemini Pro 2.5, so potential biases or failures in prompt generation are not discussed. There description of hyperparameters for Diffusion Forcing, PMWD window size/overlap ratios are not provided in the paper.

Clarity. The text is well-structured and mostly clear. The figures provide a good visualization of error accumulation and main improvements. Thekey algorithms (PMWD and Diffusion Forcing) are clearly motivated and described.

Significance. The proposed approach addresses a critical gap in long video generation, handling multi-scene scenarios with precise control along the temporal axis. The frame-level paradigm could influence video-language modeling beyond diffusion.

Originality. PMWD is a novel inference pipeline that avoids error accumulation via parallel bidirectional window denoising. The attention at frame-level and Diffusion Forcing provide new solutions to temporal misalignment in multi-scene video generation.

---

> ### Author Rebuttal · Authors · 2025-07-31
>
> ## General Response to All Reviewers:
>
> 1.
>    **The first trial of the frame-level caption method**, representing a significant attempt in dataset annotation paradigms. This approach offers unique advantages:
>
>    - First, the annotation method is simple, requiring only uniform frame sampling without considering clip, shot, or scene boundaries. This greatly reduces annotation difficulty while capturing natural transitions in the dataset, serving as a crucial technique for advancing multi-shot/scene generation scenarios.
>    - Second, unlike direct frame-level image caption strategies, our core innovation involves using state-of-the-art Vision-Language Models (VLM) to first establish global understanding of multi-shot long videos before annotating individual frames. This fundamentally differs from traditional pipelines that process and caption single clips separately, providing substantial significance.
>    - Finally, fine-grained frame-level captions can be flexibly rewritten and summarized into clip/chunk/shot/scene/video-level captions using LLM models based on specific usage scenarios. Particularly in chunk-level autoregressive inference, frame-level captions can form chunk prompts of arbitrary lengths without concerns about shot or scene boundaries—an issue that cannot be resolved using shot-level or scene-level captions directly.
> 2.
>    **This paper introduces semantic confusion in multi shot autoregressive long video scenarios**, going beyond the commonly discussed error accumulation problem. More text descriptions do not improve generation quality; instead, they cause semantic confusion during generation. This occurs because each chunk requires its own dedicated prompt rather than a global video-level prompt—a critical insight not addressed in previous autoregressive works focused primarily on simple actions and single shots. To quantify this issue, we introduce the 'confusion degree' metric.
> 3.
>    This paper also **addresses how to effectively utilize frame-level captions** through our proposed Frame-Level Attention Mechanism. By restricting each frame/chunk/shot to attend only to its corresponding prompt while maintaining global information interaction through visual self-attention, we find this approach can still achieve overall understanding.
>    This concept demonstrates remarkable foresight, as evidenced by its subsequent adoption in Seedance—the current SOTA closed-source video generation model released after our work. Seedance employs a similar mechanism where multi-modal visual-text attention operates exclusively within shots, while global information interaction through temporal self-attention (visual only) prevents inter-shot semantic confusion. This convergence confirms our approach's forward-looking and heuristic value, particularly for future advancements in multi-shot/scene long video generation.
> 4.
>    **Regarding our proposed PMWD**, we emphasize it represents just one optional inference method after frame-level training—not a core contribution of this paper. While the original PMWD shows significant advantages in addressing error accumulation, simpler diffusion-forcing sliding window inference methods could also be viable if they solve the error accumulation issue (e.g., using distribution matching techniques like DMD used in Causvid or self-forcing). Notably, PMWD's inherent parallelization capability provides distinct advantages with sufficient GPU resources, and if combined with compressed representations of infinite historical frames, offers substantial future potential.
> 5.
>    **Open Source Commitment**: We will promptly release our dataset processing code, model training/inference/evaluation code, and part of multi-shot video data with frame-level caption annotations.
> 6. **More clear implementation details:**
>     As described in line 297 and the caption of Table 1, we present a comparative analysis of models trained on the Wanx2.1 dataset using either global video-level or frame-level prompts under identical training and inference configurations. All experiments were conducted using our proposed Parallel Multi-Window Denoising (PMWD) method. A 5-second video corresponds to 21 latents, which matches the frame length used during Wanx training; therefore, we follow diffusion forcing protocols by treating these 21 latents as a sliding window unit. After each generation step, the window slides by 1/3 of its length (7 latents). For the FIFO approach, we adhere to the original paper's configuration with a slide increment of 1 latent per step. While these implementation details do not affect the conclusions, we apologize for not clearly presenting them. We will provide more detailed explanations in the main text and supplementary materials according to the reviewers' suggestions, and ensure reproducibility through open-source code.
>
> Finally, we sincerely thank all reviewers for their meticulous reviews and kindly request consideration of the heuristic contributions this paper offers to the field.
>
>
>
> ## Response To Reviewer uVip
> 1. Why only Gemini Pro 2.5 was used for prompt generation. How will the quality metrics value change with other VLM use?
>
> Unlike frame-level image captioning schemes, our approach utilizes Vision-Language Models (VLM) to first establish global understanding of the entire video before annotating individual frames. This represents a global-to-local annotation methodology that enables comprehensive global comprehension along with ID understanding and memory capabilities. These requirements place extremely high demands on VLM performance, which is why we selected the state-of-the-art Gemini Pro model available at the time. While other VLM models could potentially be used, they would likely yield inferior results due to the inherent challenges of long video understanding.
>     The cost of annotating the entire dataset using VLMs is extremely high. Therefore, we did not conduct additional VLM comparisons and only used the most advanced VLM model available. We will include comparisons of different VLMs in the supplementary materials.
>
> 2. More clear discription of Confusion Degree
>
> The traditional Video-Level Video-Text Similarity may assign higher scores even when content from different scenes are inappropriately generated in one frame, not considering temporal and semantic inaccuracies.
> A frame should only contain content specified in its corresponding prompt. If a frame contains content that should not appear in it, we define this as confusion. For example, when all content is generated in every frame and they merge together.
> In implementation, if the similarity between prompt_i of frame_i and the visual content of another frame_j is greater than the similarity between prompt_i and prompt_j, it indicates that frame_j may contain redundant content that should belong to prompt_i. This is specifically defined by the frame-level text-text similarity and frame-level text-frame similarity in Equation 3.
>
> 3. Can we use PMWD for non-overlapping frame shifts? What will be the effect of such operations?
>
> If PMWD is used for non-overlapping frame shifts, it would result in completely discontinuous video with each segment generated independently. The overlapping regions represent the information from other segments visible to each segment. While this approach shows significant potential, it needs to be combined with infinite context and compression technologies like framepack or memory techniques to improve implementation consistency in future work.
>
> 4. Computational cost evaluation
>
> For autoregressive long video generation, such as generation of 1 minute video, the computation cost and memory usage of PMWD are the same as those of sliding window inference in diffusion forcing in sequential mode inference. This is because all inference methods use the same sliding window length (21 latents): for diffusion forcing, the sliding is 7 latents per step, while for PMWD it is a 7-latent overlap, resulting in the same additional overhead. However, in parallel mode, PMWD can utilize more GPU resources to generate each sliding window in parallel, where the total generation time is inversely proportional to the resources used—a trade-off.

---

> > ### Comment · Reviewer_uVip · 2025-08-05
> >
> > Thanks for the author's reply. I appreciate your answers and including comparisons of different VLMs in the supplementary material. However, I will remain the score

---

### Decision · Program_Chairs · 2025-09-17

**Decision:**

Reject

**Comment:**

x1 accept, x2 borderline rejects, x1 reject: This paper introduces a framework for generating long, multi-scene videos that replaces the standard single-prompt method with a system using frame-by-frame text annotations. This includes a new dataset, a corresponding attention mechanism, and a parallel inference strategy to improve coherence and reduce error accumulation. The reviewers agree on the value of the frame-level dataset contribution and acknowledge it as first-of-its-kind at this scale. The weaknesses noted by reviewers include limited contribution beyond the dataset, presentation and clarity issues, lack of baseline comparisons with existing methods, and missing implementation details. The authors' follow-up responses addressed some concerns and highlighted important points, including their approach to semantic confusion in multi-scene generation and their commitment to include comparisons of different VLMs in supplementary materials. However, concerns about the core contribution and presentation remained, with reviewers finding the frame-level prompting approach insufficient as the primary contribution.

The AC acknowledges the potential value of the frame-level dataset and the authors' commitment to release code and data. However, the unresolved fundamental issues regarding limited contribution and presentation problems present significant barriers. Therefore, the AC leans not to accept this submission.